

# Role of Sb in the superconducting kagome metal CsV$_3$Sb$_5$ revealed by its anisotropic compression

Alexander A. Tsirlin[1*], Pierre Fertey[2], Brenden R. Ortiz[3,4], Berina Klis[5], Valentino Merkl[5], Martin Dressel[5], Stephen D. Wilson[4] and Ece Uykur[5†]

**1** Experimental Physics VI, Center for Electronic Correlations and Magnetism, University of Augsburg, 86159 Augsburg, Germany
**2** Synchrotron SOLEIL, L'Orme des Merisiers, F-91190 Saint-Aubin, Gif-sur-Yvette, France
**3** Materials Department, University of California Santa Barbara, Santa Barbara, CA, 93106, United States
**4** California Nanosystems Institute, University of California Santa Barbara, Santa Barbara, CA, 93106, United States
**5** 1. Physikalisches Institut, Universität Stuttgart, D-70569 Stuttgart, Germany

★ altsirlin@gmail.com, † ece.uykur@pi1.physik.uni-stuttgart.de

## Abstract

Pressure evolution of the superconducting kagome metal CsV$_3$Sb$_5$ is studied with single-crystal x-ray diffraction and density-functional band-structure calculations. A highly anisotropic compression observed up to 5 GPa is ascribed to the fast shrinkage of the Cs–Sb distances and suppression of Cs rattling motion. This prevents Sb displacements required to stabilize the three-dimensional charge-density-wave (CDW) order and elucidates the disappearance of the CDW already at 2 GPa despite only minor changes in the electronic structure of the normal state. At higher pressures, vanadium bands still change only marginally, whereas antimony bands undergo a major reconstruction caused by the gradual formation of the interlayer Sb–Sb bonds. Our results exclude pressure tuning of vanadium kagome bands as the main mechanism for the non-trivial evolution of superconductivity in real-world kagome metals. Concurrently, we establish the central role of Sb atoms in the stabilization of a three-dimensional CDW and Fermi surface reconstruction.

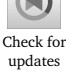

# 1  Introduction

Kagome geometry leads to a strong magnetic frustration, but also to an interesting physics in the metallic state. Here, the presence of linear bands, Dirac points, and band saddle points (van Hove singularities) in the electronic structure of a nearest-neighbor kagome metal causes multiple electronic instabilities that have been studied theoretically [1–4] but did not find their way into the lab until the recent discovery of the $AV_3Sb_5$ compound family with A = K, Rb, Cs [5]. Both charge-density wave (CDW) order [6–11] and superconductivity [12–17] observed experimentally in these compounds closely resemble theoretical findings for a simple kagome metal with the Fermi energy lying near the van Hove singularity at the filling level of $\frac{5}{12}$ [18, 19]. In $AV_3Sb_5$, the corresponding saddle points are clearly visible in the vanadium bands around the $M$ point slightly below the Fermi level (Fig. 1).

The electronic instabilities in $AV_3Sb_5$ further show strong sensitivity to the hydrostatic pressure. The superconducting transition temperature $T_c$ changes non-monotonically [20–22] and reveals a re-entrant behavior [23] that may be indeed expected in the simple kagome metal when positions of its band saddle points are tuned, and the system goes through several exotic electronic phases, including superconducting phases with unconventional pairing [1, 2, 4, 19, 24]. First experimental results suggest that pressure-tuned $AV_3Sb_5$ compounds could indeed allow an experimental access to the physics of simple kagome metals with variable positions of band saddle points relative to the Fermi level.

Here, we critically test this hypothesis by studying the crystal structure as well as electronic structure of $CsV_3Sb_5$ under hydrostatic pressure. Previous transport measurements on this compound identified the onset of CDW at $T_{CDW} = 94$ K at ambient pressure [25]. The transition temperature decreases upon compression until the CDW phase disappears around 2 GPa [20], whereas superconducting $T_c$ concomitantly increases from 2.5 K to 8 K [21–23]. On further compression, $T_c$ decreases up to pressures of about 10 GPa and then increases again, thus giving rise to a double-dome structure in the phase diagram and the apparent re-entrant behavior [23, 26] that hitherto lacks any microscopic explanation. Concurrent reports of a similar phenomenology in the K and Rb compounds [27,28] suggest that this type of pressure-induced behavior is generic for the whole family of real-world kagome metals. It is then even more intriguing to verify how vanadium kagome bands in these compounds evolve under pressure, how positions of band saddle points change, and whether a direct link to the theoretical results for kagome metals can be established.

In our work, we identify main effects that can be responsible for the suppression of CDW and re-entrant superconductivity in $AV_3Sb_5$. Compression of these materials appears to be strongly anisotropic. The Sb atoms are pushed toward each other and prompted to form strong covalent bonds. The concomitant reconstruction of the Fermi surface appears to be the main effect on the electronic structure, whereas only minor changes occur in the vanadium kagome bands. This excludes the possibility that pressure-tuned $AV_3Sb_5$ compounds allow a significant variation in the positions of band saddle points in a kagome metal, and further rules out the idea that these saddle points are mainly responsible for the pressure-induced physics. Concurrently, our results provide essential guidance on the realistic modeling of $AV_3Sb_5$ under

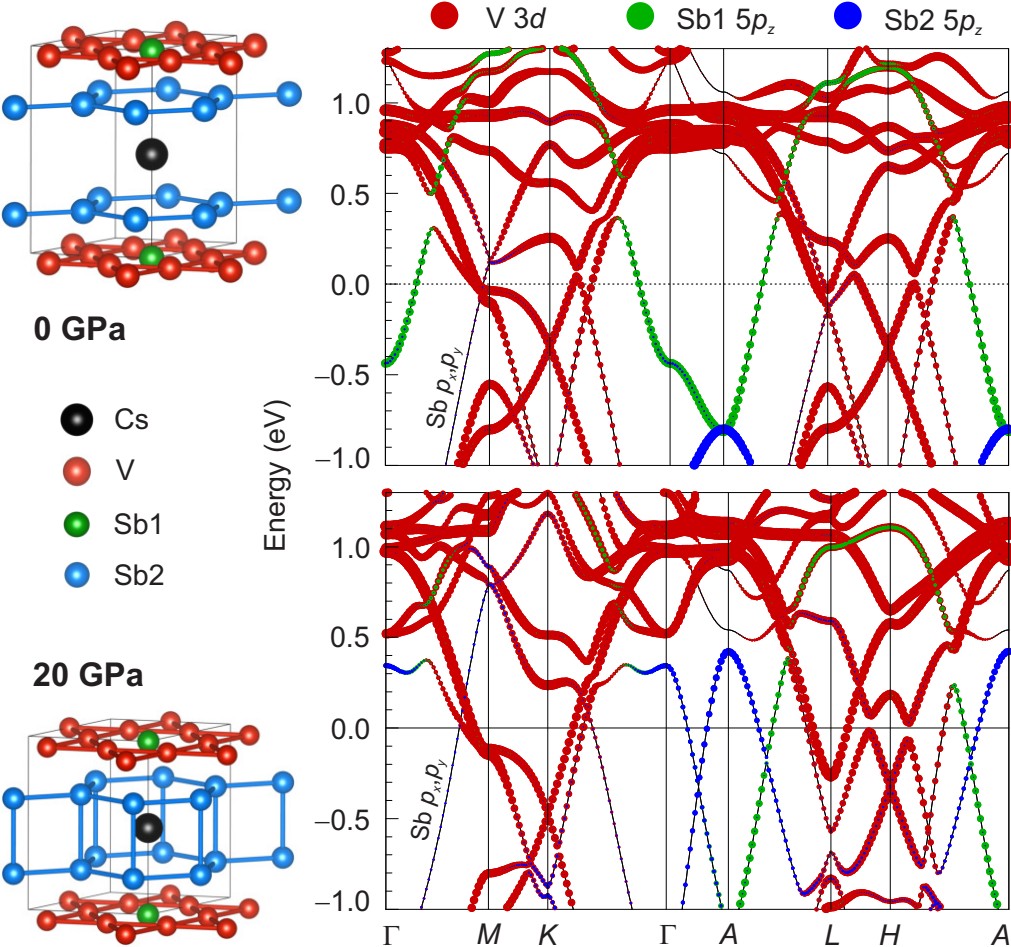

Figure 1: **Comparison of the crystal and electronic structures of CsV$_3$Sb$_5$ at 0 GPa (top) and 20 GPa (bottom).** Colored dots in the band dispersions show contributions of different atomic orbitals ("fat bands"). The remaining band along $\Gamma - M$ has predominantly Sb2 $5p_x, 5p_y$ character. VESTA program [39] was used for the crystal structure visualization.

pressure, and suggest that Sb atoms should be considered on equal footing with vanadium kagome bands.

## 2 Methods

**X-ray diffraction measurements.** Our structural study was performed on a single crystal of CsV$_3$Sb$_5$ [25] loaded into a diamond anvil cell filled with neon gas as pressure transmitting medium. Pressure values were independently calibrated using ruby luminescence and the lattice parameter of gold powder placed into the same cell. X-ray diffraction (XRD) measurements were performed at room temperature on a six-circle diffractometer of the CRISTAL beamline at SOLEIL. Diffraction data were collected with the x-ray wavelength of 0.42438 Å by 1° rotations of the cell about the $\varphi$ axis. The data were processed using Crysalis Pro [29] and corrected for the absorption in Absorb [30] by taking crystal shape into account. About 420 reflections with 70 unique reflections were used to refine 5 parameters of the CsV$_3$Sb$_5$ structure with the $P6/mmm$ symmetry: $z$-coordinate of Sb1 and isotropic displacement pa-

rameters of Cs, V, Sb1, and Sb2[1]. The refinements were performed in Jana2006 [31] against the data collected at pressures up to 20 GPa. Further compression led to a rapid deterioration of the crystal quality, such that only lattice parameters could be determined at 22 GPa.

Note that all diffraction measurements were performed at room temperature in order to assess crystal and electronic structures of $CsV_3Sb_5$ in its normal state. Electronic instabilities of the normal state would then lead to the formation of a charge-density wave upon cooling, as observed at ambient pressure. Likewise, the evolution of the normal-state band structure should be responsible for changes in the superconducting state as a function of pressure.

**DFT calculations.** Full-relativistic DFT calculations were performed in the FPLO [32] and Wien2K [33, 34] codes using Perdew-Burke-Ernzerhof (PBE) exchange-correlation potential [35] and the $24 \times 24 \times 12$ $k$-mesh that ensured convergence of the Fermi energy. Both lattice parameters and atomic positions were fixed to their experimental values determined from XRD. This approach gives more reliable and unambiguous results compared to the fully *ab initio* procedure where structural parameters are relaxed, and inaccuracies in the description of anisotropic chemical bonding by DFT functionals are included in the calculated band structure implicitly. Indeed, previous DFT studies at ambient pressure revealed significant discrepancies in the energies of the saddle points at $M$ [5, 6, 9], likely because of the different input crystal structures used in the calculations. For example, full structural relaxation results in the unit cell volume $V = 237.7 \mathring{A}^3$ and $c/a = 1.675$ with the PBE functional or $V = 240.8 \mathring{A}^3$ and $c/a = 1.715$ with the D3 correction [36], to be compared with the experimental structural parameters $V = 243.4 \mathring{A}^3$ and $c/a = 1.694$ at ambient pressure [5]. The proximity of the saddle points to the Fermi level is deemed crucial for the electronic instability and CDW formation [19, 37, 38]. Using experimental structural parameters eliminates any ambiguity in the positioning of these saddle points relative to $E_F$.

Additionally, VASP code [40, 41] was used to determine stabilization energy of the CDW state. Here, experimental lattice parameters were again fixed, while atomic positions were allowed to relax for a reliable comparison between the normal and CDW states. The VASP results have been cross-checked in FPLO for the 0 GPa crystal structure.

## 3 Results and Discussion

### 3.1 Crystal Structure

The pressure evolution of the $CsV_3Sb_5$ crystal structure is summarized in Fig. 2. The in-plane lattice parameter $a$ changes linearly, whereas the out-of-plane parameter $c$ shows a striking non-linear behavior and decreases by more than 10% at 5 GPa. At higher pressures, the evolution of $c$ also becomes linear, albeit with a higher slope than in the case of $a$. Therefore, $c/a$ is systematically reduced under pressure. No signs of symmetry lowering or a structural phase transition were observed within the pressure range of our study.

Structure refinement shows that this anisotropic compression is related exclusively to the shrinkage of the Cs–Sb distances. Their pressure evolution parallels the decrease in the $c$ lattice parameter and becomes linear above 5 GPa. In this linear regime, we determine the bond compression of 0.018 Å/GPa to be compared with 0.007 Å/GPa for the V–Sb bonds that evolve linearly starting from ambient pressure.

Remarkably, no differences between the pressure evolution of the V–Sb1 and V–Sb2 distances are observed. This identifies the $[V_3Sb_5]$ slabs as rigid structural units that are glued together by weakly bonded alkaline-metal atoms. Indeed, at ambient pressure the atomic displacement parameter (ADP) of Cs is 3 times higher than those of Sb and V, despite the fact

---

[1]See Supplemental Material for details of the data collection and sample structure refinements.

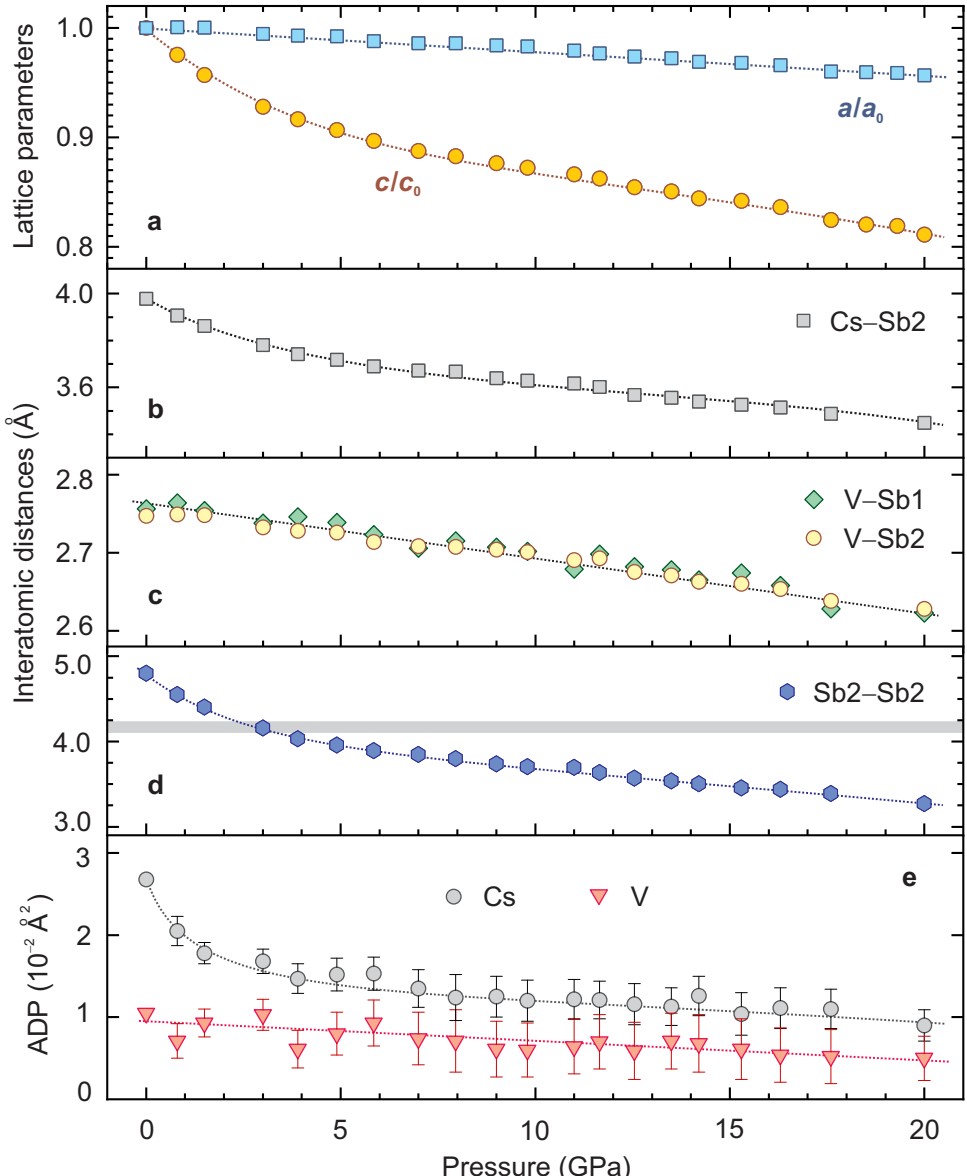

Figure 2: **Structural parameters of CsV$_3$Sb$_5$ as a function of pressure a** Lattice parameters $a$ and $c$ relative to their ambient-pressure values $a_0$ and $c_0$, respectively. **b, c, d** Interatomic distances. The shaded line in **d** shows the doubled atomic radius of Sb as the value that demarcates the formation of Sb2-Sb2 chemical bonds between the [V$_3$Sb$_5$] layers. **e** Atomic displacement parameters (ADPs) of Cs and V. The ambient-pressure values are from Ref. [5]. The error bars are from the structure refinement and smaller than the symbol size, except for the ADPs. The lines are guide-for-the-eye only.

that Cs is the heaviest atom in the structure. This abnormally high ADP indicates rattling motion expected for an atom placed into a large cavity. The rattling is suppressed around 5 GPa. Above 5 GPa, all ADPs decrease only weakly, owing to the gradual hardening of the phonon modes upon compression.

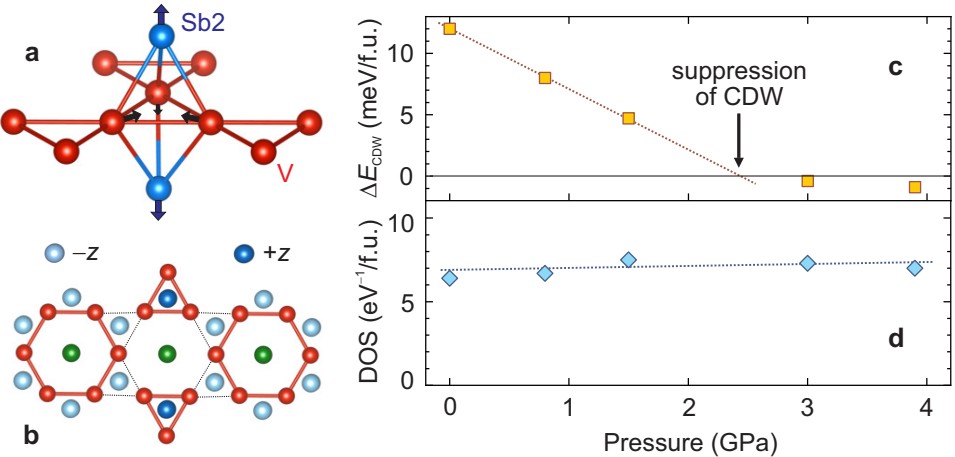

Figure 3: **Stability of the CDW state under pressure. a** Shortened V–V distances require that the adjacent Sb2 atoms move away from the kagome plane. **b** Tri-hexagonal CDW structure with opposite displacements of the Sb2 atoms depending on their proximity to the short V–V bonds ($+z$ stands for displacements away from the kagome planes). **d** Stabilization energy of the tri-hexagonal CDW state compared to the undistorted structure. **d** Density of states at $E_F$ for the undistorted structure does not change across the pressure range where CDW is suppressed.

## 3.2 CDW order

We now turn to the CDW state and calculate its stabilization energy as a function of pressure (Fig. 3c) using tri-hexagonal (inverse star-of-David) type of distortion as the most plausible model of the CDW [8, 9, 42–44]. Note that similar arguments apply to any other type of distortion that involves the modulation of V–V bonds. The stabilization energy of the CDW decreases with pressure and becomes negative above $P_{CDW} \simeq 2.5$ GPa in excellent agreement with the experiment [20].

The pressure increase leads to a suppression of the CDW but does not cause any significant changes in the band structure of the normal state. The saddle points around $M$ associated with the CDW instability do not shift in this pressure range at all (Fig. 4a). The density of states at the Fermi level remains as high as at ambient pressure (Fig. 3d). These observations do not exclude the electronic origin of the CDW, but witness that electronic instability alone is not sufficient to induce the distortion.

The reduction in $T_{CDW}$ under pressure can be understood by inspecting structural changes in the CDW state. The shortening of all V–V distances in a triangle leads to an overbonding, which should be compensated by shifting the respective Sb2 atoms along the $c$ axis away from the kagome plane (Fig. 3a). Conversely, the shortening of one V–V distance and the elongation of the other two is compensated by shifting the Sb2 atom toward the kagome plane (Fig. 3b). Such displacements of Sb become increasingly less favorable as the crystal shrinks along the $c$ direction and the Cs–Sb2 distances decrease.

The good agreement between our estimated $P_{CDW}$ and the experimental value of about 2 GPa [20] shows that lattice energy associated with the Sb2 displacements plays an important role in stabilizing the three-dimensional CDW state. This is further confirmed by the fact that tri-hexagonal distortion of the kagome plane can be obtained also above $P_{CDW}$, but the resulting structure is only a local energy minimum lying higher in energy than the parent undistorted one (Fig. 3c). Indeed, relative displacements of the Sb2 atoms in the tri-hexagonal CDW state decrease from 0.06 Å at ambient pressure to only 0.02 Å at 3 GPa.

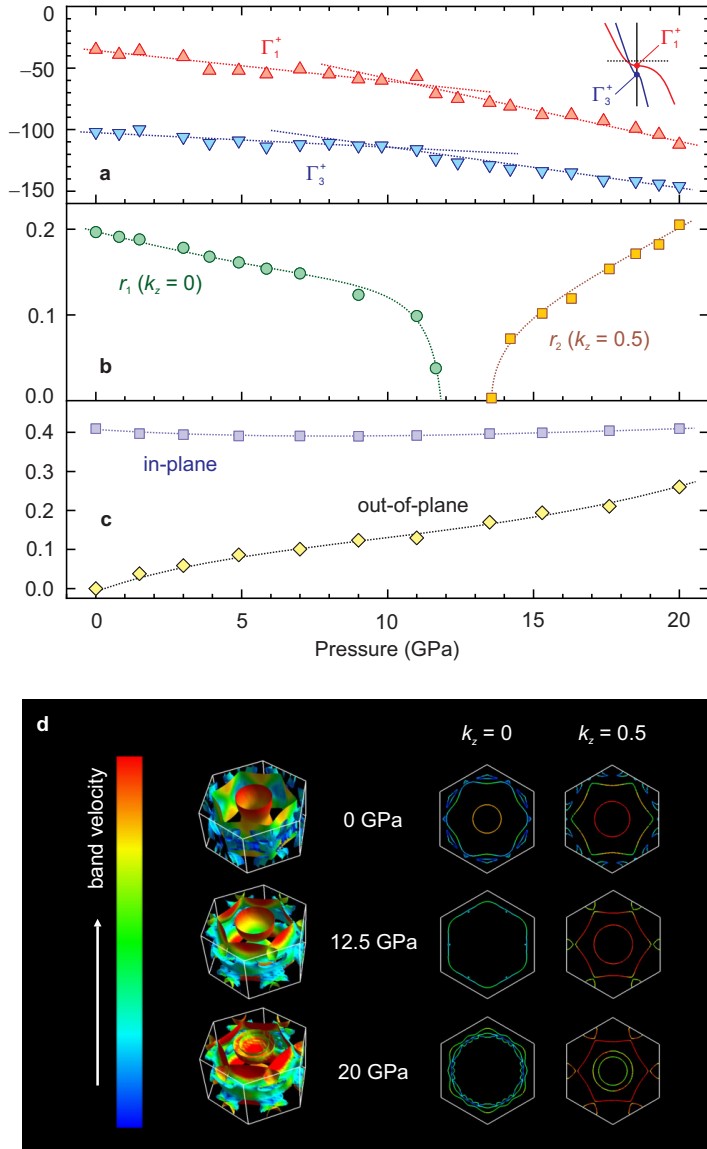

Figure 4: **Pressure evolution of the band structure. a** Energies of saddle points at $M$ relative to $E_F$. **b** Size of the Sb $p_z$ Fermi surface (FS) pockets around $\Gamma$ ($k_z = 0$) and $A$ ($k_z = 0.5$) points. **c** Electron hopping for the in-plane and out-of-plane Sb2– Sb2 bonds ($\pi$-orbitals are chosen in both cases). **d** Representative Fermi surfaces, with the color scheme showing band velocities. The lines are guide-for-the-eye only.

## 3.3 Band Structure

In order to understand pressure-induced changes in the electronic structure, we first analyze energy bands in the vicinity of $E_F$ at ambient pressure (Fig. 1). Apart from vanadium bands of a kagome metal, several bands of purely Sb $5p$ origin are observed. They arise from the covalent Sb2–Sb2 bonds within graphite-like layers, as well as the bonds between Sb1 and Sb2. The respective bond distances of 3.17 Å and 3.89 Å are much shorter than 4.12 Å, the doubled atomic radius of Sb.

A comparison of the band structures calculated at 0 and 20 GPa (Fig. 1) reveals that V bands are only slightly broadened under pressure, whereas Sb bands undergo a major reconstruction. These changes are tracked, respectively, by the energies of the saddle points for the V bands, and by the radii of the Fermi surface pockets for the Sb $p_z$ bands (Fig. 4). Note that we

disregard the Sb band formed by the $p_x$ and $p_y$ orbitals, because its Fermi surface changes only marginally, contrary to the $p_z$ channel where the Sb2 band, which is capped at $-0.8$ eV at 0 GPa, goes well above $E_F$ at 20 GPa and pushes down a significant fraction of the Sb1 states that were predominant around $E_F$ at ambient pressure.

Cylindrical Fermi surface (FS) around $\Gamma$ originates from these Sb1 $p_z$ states and gradually shrinks under pressure (Fig. 4b). It disappears around 12 GPa shortly after the end of the first superconducting dome. Around 13.5 GPa, the FS pocket due to the Sb2 $p_z$ states appears around $A$ and precedes the second superconducting dome. Compared to this major reconstruction of the Sb $p_z$ bands, the changes in the V bands are minor. The saddle points shift to lower energies as the width of the V bands increases (Fig. 4a). The change in the slope of this pressure dependence around 11 GPa can be ascribed to the evolution of the Sb $p_z$ FS, because the population and depopulation of the respective bands pins the Fermi level, thus affecting the positions of the saddle points relative to it. Other than that, the V bands retain their dispersions imposed by the kagome framework, and the associated Fermi surfaces remain in place, except for a small pocket along $L - H$ that disappears above 10 GPa.

The changes in the Sb $p_z$ bands are elucidated by pressure evolution of the hopping parameters extracted using Wannier fits. In Fig. 4c, we compare $\pi$-type Sb2–Sb2 hoppings within graphite-like nets in the $ab$-plane and perpendicular to these nets along the $c$ axis. The out-of-plane hopping, which is essentially absent at ambient pressure, becomes comparable to the in-plane hopping at 20 GPa. This reflects the formation of the new covalent bonds between the $V_3Sb_5$ slabs while the respective Sb2–Sb2 distance shrinks from 4.8 Å at 0 GPa to 3.27 Å at 20 GPa (Fig. 1). The interlayer Sb2-Sb2 bond can be formally tracked starting from 2-3 GPa where the respective distance goes below the doubled atomic radius of Sb. Therefore, the 2D-3D crossover starts as soon as the CDW is suppressed, although the effect of this crossover on the band structure becomes tangible only above 10 GPa where re-entrant superconductivity has been observed.

## Conclusion and Outlook

Kagome bands with their saddle points and Dirac points are so far deemed the backbone of the $AV_3Sb_5$ electronic structure and the origin of the interesting physics discovered therein. We show that these kagome bands dominated by V $3d$ states are, in fact, only one important ingredient. The Sb $5p$ states and the Sb atoms become crucial when pressure evolution of $CsV_3Sb_5$ is considered or the stability of the CDW state is concerned. A CDW can be envisaged in a compressed structure too, but it is no longer a global energy minimum, because Sb2 displacements stabilizing the three-dimensional CDW are suppressed. Recent spectroscopy experiments hint at the CDW gap opening around $M$ [7, 45], in accord with theoretical studies that discuss saddle points of the kagome bands as the microscopic reason for the CDW formation [18, 38]. However, it is Sb atoms that decide whether the CDW state eventually forms or not.

Turning to pressures beyond the CDW state, we note that several changes in the crystal and electronic structures accompany the gradual suppression of superconductivity between 2 and 10 GPa. The rattling motion of Cs is damped (Fig. 2d), and a small FS pocket related to the V $3d$ bands at $L - H$ disappears (Fig. 1). However, the main changes occur in the Sb $5p$ bands. The disappearance of the Sb1 FS around $\Gamma$ is followed by the formation of the Sb2 FS around $A$ (Fig. 4b). This evolution mimics the suppression and re-entrance of superconductivity observed in transport measurements [23, 26]. At 12.5 GPa, in between the two superconducting domes, the number of FS pockets and the overall area of the FS is the lowest compared to both ambient pressure and 20 GPa (Fig. 4d). These changes suggest that

the FS size and especially the presence of Sb pockets may be crucial for the superconductivity in $CsV_3Sb_5$. Such observations corroborate recent ideas that the superconductivity in $CsV_3Sb_5$ is unconventional in nature [19, 43].

On the structural level, the pressure evolution of $CsV_3Sb_5$ is governed by the gradual formation of the covalent Sb2–Sb2 bonds that link the $V_3Sb_5$ slabs into a three-dimensional network (Fig. 1). This behavior is remarkably similar to some of the Fe-based superconductors [46], where collapsed crystal structures are stabilized by pressure. The important difference of $CsV_3Sb_5$ is that such structural changes are not accompanied by an abrupt phase transition. The anisotropic compression is gradual and reversible. The proclivity of $CsV_3Sb_5$ for the compression along $c$ and major changes in the band structure caused by this compression suggest that elastic strain tuning may be a promising strategy for this material. Moreover, its potential superelasticity [47] can also be worth exploring. The scenario of anisotropic compression and concomitant changes in the Sb Fermi surface should be also applicable to the sister compounds $KV_3Sb_5$ and $RbV_3Sb_5$ that show a similar phenomenology according to the recent resistivity measurements [27, 28].

# Acknowledgements

We are grateful to Gabriele Untereiner for preparing single crystals for the XRD experiment as well as Alain Polian for preparing the pressure cells for measurements. We also acknowledge SOLEIL for providing the beamtime.

**Funding information**  S.D.W. and B.R.O. gratefully acknowledge support via the UC Santa Barbara NSF Quantum Foundry funded via the Q-AMASE-i program under award DMR-1906325. B.R.O. also acknowledges support from the California NanoSystems Institute through the Elings fellowship program. The work has been supported by the Deutsche Forschungsgemeinschaft (DFG) via DR228/51-1 and UY63/2-1. E.U. acknowledges the European Social Fund and the Baden-Württemberg Stiftung for the financial support of this research project by the Eliteprogramme.

# A   Supplemental Material

Table 1: Details of data collection and refined structural parameters for $CsV_3Sb_5$ at 3.0 GPa.

| $a = b = 5.4651(2)$ Å,   $c = 8.637(7)$ Å |
|---|
| $V = 223.4(2)$ Å$^3$ |
| $P6/mmm$ |
| $\lambda = 0.42438$ Å |
| $\theta_{\min} = 2.57°$, $\theta_{\max} = 18.52°$ |
| $-6 \leq h \leq 5$,   $-7 \leq k \leq 8$,   $-3 \leq l \leq 3$ |
| $R_{\mathrm{int}} = 0.080$, $R_{I>3\sigma(I)} = 0.061$, $wR_{I>3\sigma(I)} = 0.065$ |

| Atom | $x/a$ | $y/b$ | $z/c$ | $U_{\mathrm{iso}}$ (Å$^2$) |
|---|---|---|---|---|
| Cs | 0 | 0 | 0 | 0.0168(15) |
| V | 0.5 | 0.5 | 0.5 | 0.0103(19) |
| Sb1 | 0 | 0 | 0.5 | 0.0098(15) |
| Sb2 | $\frac{2}{3}$ | $\frac{1}{3}$ | 0.7591(11) | 0.0130(10) |

Table 2: Details of data collection and refined structural parameters for $CsV_3Sb_5$ at 20.0 GPa.

| $a = b = 5.2562(4)\,\text{Å}, \quad c = 7.552(9)\,\text{Å}$ |
|:---:|
| $V = 180.7(2)\,\text{Å}^3$ |
| $P6/mmm$ |
| $\lambda = 0.42438\,\text{Å}$ |
| $\theta_{\min} = 2.67°, \ \theta_{\max} = 18.11°$ |
| $-7 \leq h \leq 7, \quad -7 \leq k \leq 7, \quad -3 \leq l \leq 3$ |
| $R_{\text{int}} = 0.068, R_{I>3\sigma(I)} = 0.097, wR_{I>3\sigma(I)} = 0.093$ |

| Atom | $x/a$ | $y/b$ | $z/c$ | $U_{\text{iso}}$ (Å$^2$) |
|:---:|:---:|:---:|:---:|:---:|
| Cs | 0 | 0 | 0 | 0.0090(19) |
| V | 0.5 | 0.5 | 0.5 | 0.0050(30) |
| Sb1 | 0 | 0 | 0.5 | 0.0110(20) |
| Sb2 | $\frac{2}{3}$ | $\frac{1}{3}$ | 0.7833(19) | 0.0117(15) |

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
