# Peer review of "Role of Sb in the superconducting kagome metal CsV$_3$Sb$_5$ revealed by its anisotropic compression"

_SciPost Physics, doi:SciPost Phys. 12, 049 (2022)_

## Round 2 · Referee Report · Anonymous (Referee 1) · 2021-11-29

Strengths

1- Comparatively compact paper on comparison between high pressure structure experiment and bandstructure calculation 2- Well written, clear message

Weaknesses

1- Findings of limited relevance

Report

Authors report their findings regarding the structural response of CsV3Sb5 on high hydrostatic pressure. The results are discussed in terms of the bandstructure calculated as function of pressure. The authors find good agreement between anomalies in their structural study and the disappearance of a charge densitiy/reappearance of superconductivity. Very straightforward message, although in the end the conclusion that standard bandstructure calculations work, and that you need to consider all atoms in a material if you to really understand its properties. Still, paper meets criteria regarding scientific validity and relevance and should essentially be published as is.

Requested changes

1- The authors should note the temperature, at which the high pressure experiments have been performed. 2- The authors should note and possibly discuss if the supposedly high (room) temperature affects material properties. Are there low temperature structural data?

---

## Round 2 · Referee Report · Anonymous (Referee 2) · 2021-12-9

Strengths

  1. A potentially useful contribution to the growing field of Kagome materials.
  2. Detailed discussion of DFT results

Weaknesses

A rather technical work, which in present form is difficult to read for a non-specialist.

Report

Please note that I am a condensed matter theorist, and not an expert in DFT, so it is hard for me to judge the validity or importance of the paper for the DFT community.

In terms of the relevance to condensed matter theory and experiment, the results presented in the paper may be useful, given that there is a general interest in Kagome materials.

However, the main outcome of the paper regarding the importance of the contribution of Sb atoms to the band structure, etc. seems to be rather limited, and this point alone may not be sufficient to pass the threshold in terms of acceptance criteria of this journal.

general remarks:

The authors say: "Therefore, chemical pressure has only a minor effect on the low-temperature behavior. In contrast, changes induced by hydrostatic pressure are drastic." Would it be possible to add a discussion on why there is such a difference in the effect of chemical pressure vs hydrostatic pressure, and perhaps give other examples of materials which show similar behaviour.

Requested changes

In my opinion the paper would benefit from including an introduction on the physics of Kagome metals, more detailed discussion of the results in terms of their comparison with other materials, e.g. Fe-based superconductors (mentioned in the paper), and in general putting the results into broader perspective.

---

## Round 3 · Author Response

Hereby we resubmit our manuscript and briefly respond to the referees' comments.

Referee 1:

1- The authors should note the temperature, at which the high pressure experiments have been performed.

Diffraction experiments were performed at room temperature, as clearly stated in the beginning of Section 2. We believe that no further changes are needed with respect to this criticism.

2- The authors should note and possibly discuss if the supposedly high (room) temperature affects material properties. Are there low temperature structural data?

We thank the Referee for this question. The idea here is that any low-temperature effects (charge-density wave, superconductivity) should be driven by and, thus, interpreted using instabilities of the electronic structure in the normal state. Therefore, we perform our structural studies at room temperature and introduce the resulting crystal structures into DFT calculations. We have commented on this point by adding an extra paragraph to Sec. 2 right after the technical description of the XRD experiments.

Referee 2:

In my opinion the paper would benefit from including an introduction on the physics of Kagome metals, more detailed discussion of the results in terms of their comparison with other materials, e.g. Fe-based superconductors (mentioned in the paper), and in general putting the results into broader perspective.

We thank the Referee for this comment. The Introduction (Sec. I) has been revised accordingly, and a broader perspective on the kagome metals has been provided, especially in the first paragraph. We prefer to avoid an extensive discussion of Fe-based superconductors, as in our opinion the analogy is more chemical (pressure-induced formation of chemical bonds between the pnictogen atoms) than intrinsic. On the level of the electronic structure, there seems to be very little similarity between the Fe-based superconductors and kagome metals.

The referees further raised the point of "limited relevance" that was echoed in the editor's request to explain how the manuscript meets the acceptance criteria of the journal. We comment on this aspect in the following.

The AV3Sb5 compounds received a lot of attention over the last months as the best available experimental realizations of a simple kagome metal. A unique aspect of these compounds is the dispersions of their vanadium bands that are strongly reminiscent of the nearest-neighbor tight-binding model on the kagome lattice. The AV3Sb5 compounds further show a gamut of experimental features expected theoretically from the kagome metal, most notably, the charge-density-wave order and superconductivity. Hydrostatic pressure is a crucial tuning parameter that affects different electronic instabilities, suppresses charge-density wave, and leads to the double superconducting dome. A common intuition, especially in the theory community, is that all these aspects can (and should) be explained by the evolution of the kagome bands, in particular band saddle points with respect to the Fermi level. This evolution is already known from theory, and it is not unnatural to expect that the same physics appears in the real material. Nevertheless, an experimental, material-specific check of such theoretical expectations is, of course, essential.

We perform this check and uncover the crystal structure, as well as electronic structure of CsV3Sb5 under pressure for the first time. We identify acute deviations from the kagome scenario and basically disprove everything that theoretical intuition tells one about the AV3Sb5 materials under pressure. Band saddle points are very weakly affected by hydrostatic pressure. Their marginal shift in energy explains neither the suppression of the charge-density wave nor the re-entrant behavior of superconductivity. On the other hand, Sb atoms -- a completely different and hitherto overlooked part of the material -- prove crucial for all pressure-induced effects in CsV3Sb5.

In this context, we believe that our manuscript meets the following acceptance criteria of SciPost Physics:

  1. Present a breakthrough on a previously-identified and long-standing research stumbling block.

The absence of simple material realizations for nearest-neighbor kagome metals has been a long-standing block for an experimental test of theories developed for kagome metals over the last decade. While this block has been seemingly removed with the discovery of AV3Sb5, it is crucial to appreciate the limitations of the kagome scenario when real materials are concerned. Our work does exactly that and serves as an important caveat, but also the guidance for the realistic theoretical description of real-world kagome metals.

  1. Open a new pathway in an existing or a new research direction, with clear potential for multipronged follow-up work.

Our work sets the stage for a microscopic description of AV3Sb5 under pressure. Every study of pressure-induced behavior, both experimental and theoretical, has to resort to changes in the crystal and electronic structures of the material. It is exactly the information that our manuscript reports.

  1. Provide a novel and synergetic link between different research areas.

Our work boasts a useful synergy of state-of-the-art high-pressure crystallography and ab initio analysis of the electronic structure. To the best of our knowledge, such combined studies have never been performed for kagome metals, and remain very rare for electronic/correlated materials in general. Our work sets the working example demonstrating how this powerful combination delivers crucial insights into an electronic/correlated material of current interest. The methodological implications can be two-fold. On the one hand, high-pressure crystallographers, who are traditionally concentrated on mineral samples of geological relevance, will be prompted to pay more attention to the structural evolution of electronic/correlated materials. On the other hand, condensed-matter experts will be prompted to use experimental crystallographic information, rather than guess pressure-induced structural changes from intuition or DFT.

As a completely different way of looking at the relevance of our article, we would like to point out that after it appeared on arXiv and has been under consideration in SciPost Physics (with only one round of review) for as long as 5 months, it has been cited in several other papers, namely:

PRB 103, 224513 (2021) PRB 104, 195130 (2021) PRB 104, 144506 (2021) PRB 104, 205129 (2021) arXiv:2108.09434 arXiv:2109.06809 arXiv:2110.09056 arXiv:2110.10171 arXiv:2110.10553 arXiv:2111.09342 Perspective in Nature Physics (doi:10.1038/s41567-021-01404-y)

This list is probably not exhaustive, as new preprints appear almost weekly. In our opinion, this level of citations and community's interest indicates high significance of the topic and rules out any speculations on the limited relevance of our work.

---

## Round 3 · List of Changes

1. Introduction has been revised.
2. One extra paragraph explaining the relevance of the room-temperature crystal structure has been added to Sec. II

---

## Editorial Decision

published